# Molecular characterization and drug resistance patterns of *Mycobacterium tuberculosis* complex in extrapulmonary tuberculosis patients in Addis Ababa, Ethiopia

**Getu Diriba**[1,2]*, **Abebaw Kebede**[1,3], **Habteyes Hailu Tola**[1], **Bazezew Yenew**[1], **Shewki Moga**[1], **Desalegn Addise**[1], **Ayinalem Alemu**[1], **Zemedu Mohammed**[1], **Muluwork Getahun**[1], **Mengistu Fantahun**[4], **Mengistu Tadesse**[1], **Biniyam Dagne**[1], **Misikir Amare**[1], **Gebeyehu Assefa**[5], **Dessie Abera**[2], **Kassu Desta**[2]

**1** Ethiopian Public Health Institute, Addis Ababa, Ethiopia, **2** Department of Medical Laboratory Sciences, College of Health Sciences, Addis Ababa University, Addis Ababa, Ethiopia, **3** Department of Microbial, Cellular and Molecular Biology, College of Natural and Computational Sciences, Addis Ababa University, Addis Ababa, Ethiopia, **4** St. Paul's Hospital Millennium Medical College, Addis Ababa, Ethiopia, **5** Armauer Hansen Research Institute, Addis Ababa, Ethiopia

* getud2020@gmail.com

**Data Availability Statement:** All relevant data are within the paper and its Supporting information files.

## Abstract

### Background

Molecular characterization of *Mycobacterium tuberculosis* (MTB) is important to understand the pathogenesis, diagnosis, treatment, and prevention of tuberculosis (TB). However, there is limited information on molecular characteristics and drug-resistant patterns of MTB in patients with extra-pulmonary tuberculosis (EPTB) in Ethiopia. Thus, this study aimed to determine the molecular characteristics and drug resistance patterns of MTB in patients with EPTB in Addis Ababa, Ethiopia.

### Methods

This study was conducted on frozen stored isolates of EPTB survey conducted in Addis Ababa, Ethiopia. A drug susceptibility test was performed using BACTEC-MGIT 960. Species and strain identification were performed using the Geno-Type MTBC and spoligotyping technique, respectively. Data were entered into the MIRU-VNTR*plus* database to assess the spoligotype patterns of MTB. Analysis was performed using SPSS version 23, and participants' characteristics were presented by numbers and proportions.

### Results

Of 151 MTB isolates, 29 (19.2%) were resistant to at least one drug. The highest proportion of isolates was resistant to Isoniazid (14.6%) and Pyrazinamide (14.6%). Nine percent of isolates had multidrug-resistant TB (MDR-TB), and 21.4% of them had pre-extensively drug-resistant TB (pre-XDR-TB). Among the 151 MTB isolates characterized by spoligotyping, 142 (94.6%) had known patterns, while 9 (6.0%) isolates were not matched with the

**Funding:** The authors received no specific funding for this work.

**Competing interests:** The authors have declared that no competing interests exist.

**Abbreviations:** AMK, Amikacin; CAP, Capreomycin; CSF, Cerebrospinal fluid; DST, Drug Susceptibility Testing; EMB, Ethambutol; EPHI, Ethiopian Public Health Institute; EPTB, Extrapulmonary Tuberculosis; ETH, Ethionamide; HIV, Human Immunodeficiency Virus; INH, Isoniazid; KAN, Kanamycin; MDR-TB, Multidrug Resistant Tuberculosis; MGIT, Mycobacterium Growth Indicator Tube; MOX, Moxifloxacin; MTB, *Mycobacterium tuberculosis*; OFX, Ofloxacine; PTB, Pulmonary Tuberculosis; PZA, Pyrazinamide; RIF, Rifampicin; RR, Rifampicin Resistance; SIRE, Streptomycin Isoniazid Rifampicin Ethambutol; SPSS, Statistical Package for Social Scientist; STM, Streptomycin; TB, Tuberculosis; VIO, Viomycin; WHO, World Health Organization.

MIRU-VNTR*plus* spoligotype database. Of the isolates which had known patterns, 2% was *M.bovis* while 98% *M. tuberculosis*. Forty-one different spoligotype patterns were identified. The most frequently identified SpolDB4 (SIT) wereSIT149 (21.2%), SIT53 (14.6%) and SIT26 (9.6%). The predominant genotypes identified were T (53.6%), Central Asia Strain (19.2%) and Haarlem (9.9%).

## Conclusion

The present study showed a high proportion of MDR-TB and pre-XDR-TB among EPTB patients. The strains were mostly grouped into SIT149, SIT53, and SIT26. The T family lineage was the most prevalent genotype. MDR-TB and pre-XDR-TB prevention is required to combat these strains in EPTB. A large scale study is required to describe the molecular characteristics and drug resistance patterns of MTB isolates in EPTB patients.

## Introduction

Tuberculosis (TB) continues to be the major public health problem and it is the top cause of death from a single infectious disease [1]. One-third of the world's population has been estimated to be infected by *Mycobacterium tuberculosis*. Based on the 2019 Global TB Report, 10.0 million new TB cases are estimated to occur. Ethiopia is among the 30 high TB, TB/HIV, and MDR-TB burden countries. World Health Organization (WHO) estimates TB incidence to be 151 per 100, 000 population in Ethiopia. Furthermore, the rate of RR/MDR-TB was 0.71% in new and 16% in previously treated cases [2]. There are two types of TB based on the anatomical sites where the disease is manifested. These are pulmonary TB (PTB) and extra-pulmonary TB (EPTB) [3]. Extrapulmonary TB is an important clinical problem as it accounts for about 15–20% of TB burden, and its prevalence is higher in patients co-infected with HIV [4–7].

Molecular characterization and drug susceptibility testing of *Mycobacterium tuberculosis* Complex (MTBC) strains are important to understand the transmission dynamics and drug resistance pattern. Various types of molecular typing techniques based on deoxyribonucleic acid (DNA) fingerprints have been developed and used to diagnose and characterize MTBC [8]. Extrapulmonary TB specimens have several inhibitors (include host proteins, blood, and eukaryotic DNA) that compromise the quality of PCR amplification leading to low sensitivity and false-negative results [9]. However, in resource-limited settings, spoligotyping is an important tool to analyze the distribution of various MTB genotype strains [10]. It is a PCR-based method for the detection and typing of the MTBC based on polymorphism in direct repeat locus of mycobacterial chromosome [10]. It offers important advantages such as the genotyping of strains from clinical samples that show a unique hybridization pattern [11]. Genotyping is the most valuable method to understand the MTB strains that are circulating in a community [12]. It is also the most important diagnostic method in the identification of MTB strains.

There are a few studies have been reported on the genetic variability and drug resistance patterns of MTB in patients with EPTB in Addis Ababa, Ethiopia [13, 14]. Besides, there is no strong EPTB surveillance data on genetic diversity and drug resistance patterns of MTBC that causes EPTB in Ethiopia [15]. Information from this study could play a crucial role to provide an overview of the genetic variation of MTBC from EPTB. Therefore, this study aimed to

determine the molecular characterization and drug resistance patterns of MTBC circulating from EPTB in Addis Ababa, Ethiopia.

## Materials and methods

### Study settings

A cross-sectional study was conducted in 151 culture-positive EPTB patients to determine the molecular characteristics and drug resistance patterns of MTB among culture-confirmed EPTB patients' specimens in Addis Ababa, Ethiopia. Addis Ababa is the capital city of Ethiopia which covers an area of 527 square kilometers with a total population of 3,384,569 [16]. The current study was conducted on clinical isolates of MTBC from two previous EPTB studies conducted in Addis Ababa, Ethiopia. One of the survey data included in the current study which was conducted on 152 patients with EPTB is published [17], while the second survey which is not published enrolled 778 patients. Sixty-eight EPTB presumptive patients were MTBC positive out of 152 from the first survey, whereas 85 were positive from 778 patients from the second survey. In total, 153(16.5%) MTBC isolates were obtained from a total of 930 patients from the two previous studies. Since two isolates were not recovered by sub-culturing, 151 MTBC isolates were used in the present study. In both studies, patients were enrolled consecutively upon arrival.

### Drug susceptibility testing

**First-line phenotypic drug susceptibility test.** Five first-line drugs such as streptomycin (STM), isoniazid (INH), rifampicin (RIF) ethambutol (EMB), and pyrazinamide (PZA) were tested using Mycobacterium Growth Indicator Tube 960 (MGIT 960) system. The drug susceptibility test (DST) was performed by Antibiotic Susceptibility Testing (AST) set with the proportional method recommended by the WHO [19]. The concentrations of the drugs in media were: STM 1.0$\mu$g/ml, INH 0.1$\mu$g/ml, RIF 1.0$\mu$g/ml, EMB 5$\mu$g/ml and PZA 100$\mu$g/ml. A growth tube was used for comparison. The bacterial inocula were diluted to 1:100 before inoculation into the growth control tube and 0.5 mL bacterial suspension was added into the growth control tube [18]. The inoculated tubes were incubated in the MGIT 960 system and monitored every hour for an increase in fluorescence. For SIRE sensitivity MGIT 960 tubes were incubated for a maximum of 13 days and 21 days for PZA.

**Second-line phenotypic drug susceptibility test.** Second-line DST was performed for all MDR-TB isolates using MGIT 960 systems. All liquid MGIT-positive MTB culture within 1 to 5 days were used for second-line DST. 800μl SIRE supplement and 100μl working drug solution were added into the MGIT tube which contained 7ml modified Middlebrook 7H9. A working solution of each drug was prepared at the concentration level of ofloxacin (OFX) 2.0μg/ml, capreomycin (CAP) 1.25μg/ml, amikacin (AMK) 1.0μg/ml, kanamycin (KAN) 2.5μg/ml, moxifloxacin (MOX) 2.5μg/mland ethionamide (ETH) 2.5 μg/ml based on the manufacturer recommendations [19, 20].

**Species and strain identification.** Geno-Type MTBC is a qualitative in-vitro test performed from cultured materials for the identification of species or strains belonging to the MTBCwhich include *M.tuberculosis/M.canettii*, *M.bovis*, *M.africanum*, *M.microti*, *Subspecies of bovis*, *M.caprae*, *M.bovis BCG* [21]. Spoligotyping to identify the species and strains was performed based on the method described elsewhere. A PCR-based amplification of the Direct Repeat (DR) region of the isolate was performed using oligonucleotide primers derived from the DR sequence. The amplified product was hybridized followed by subsequent membrane washing processes. Known strains of BCG and H37Rv were used as positive controls, and molecular grade water was used as a negative control. Hybridized DNA was detected by the

enhanced chemiluminescence method. The presence or absence of a spacer was used as the basis of the interpretation of the result. The whole procedure was performed as described by Kamerbeek et al. [22].

### Data analysis

The spoligotypes obtained from the laboratory result were entered into the MIRU-VNTR*plus* and compared with the existing Spoligotype International Type number. Data were double entered into SPSS version 23. Genotypic, phenotypic, and demographic data were described by number and percentage. The level of statistical significance was set at a p-value $\leq 0.05$.

### Ethical consideration and consent

This study obtained ethical approval from the Department of Medical Laboratory Sciences Research and Ethics Review Committee, Addis Ababa University. Stored isolates collected from patients that had provided informed consent in a previous study were used in the study and it was not feasible to trace and seek additional informed consent from the study participants. To ensure confidentiality, access to the results and documents were kept in a locked area. Information that identifies individual participants was not used in this study.

## Results

### Demographic and clinical characteristics of study participants

Of 151 EPTB isolates included in this study, eighty-two (54.3%) were from male patients. The mean age was 32.3 (±17.3 SD) years and 89 (59%) of the patients were in the age group of 15 to 39 years. Of the total enrolled patients, 99 (65.6%) were tuberculosis lymphadenitis, while 32 (12.2%) pleural tuberculosis [Table 1].

### First-line phenotypic drug sensitivity test

One hundred fifty-one isolates for which phenotypic DST was performed, 122 (80.8%) were sensitive to all first-line drugs (STM, INH, RIF EMB, and PZA). However, 29(19.2%) of isolates were resistant to at least one or more drugs. INH and PZA resistance proportion was the same (22, 14.6%) [Table 2]. The proportion of MDR-TB was 14 (9.3%), which included 3 of 14 (21.4%) previously treated and 11 of 137 (8%) newly treated cases [Table 2].

### Second-line phenotypic drug sensitivity test

Second-line drugs (AMK, CAP, ETH, KAN, MOX, and OFX) DST was conducted for a total of 14 (9.3%) MDR isolates. Of 14 isolates 2 (14.3%) were resistant to MOX, while 1 (7.1%) to CAP. Seven percent of isolates were resistant to ETH, whereas 2 (14.3%) resistant to OFX. Of the 14 MDR-TB isolates, three (21.4%) were pre-XDR-TB [Table 3]. One of the three (33.3%) previously treated and two of eleven (18.2%) new cases were found to be pre-XDR cases. XDR-TB cases were not observed in the present study.

### Species identification of *M. tuberculosis* complex

Of 151 isolates, 148 (98.0%) were identified as *M. tuberculosis* while three (2.0%) were *M. bovis*.

**Table 1. Demographic and clinical characteristics of EPTB patients in Addis Ababa Ethiopia (n = 151), 2020.**

| Characteristics | | Number | Percentage (%) |
|---|---|---|---|
| **Age (in year)** | | | |
| | <15 | 16 | 10.6 |
| | 15–39 | 89 | 58.9 |
| | 40–59 | 33 | 21.9 |
| | >60 | 13 | 8.6 |
| **Gender** | | | |
| | Female | 69 | 45.7 |
| | Male | 82 | 54.3 |
| **HIV status** | | | |
| | Non-reactive | 96 | 63.6 |
| | Reactive | 34 | 22.5 |
| | Unknown | 21 | 13.9 |
| **Marital status** | | | |
| | Married | 98 | 64.9 |
| | Single | 44 | 29.1 |
| | Separated | 9 | 6.0 |
| **Educational status** | | | |
| | Illiterate | 42 | 27.8 |
| | Primary School | 77 | 51.0 |
| | Secondary School | 24 | 15.9 |
| | Higher Education | 8 | 5.3 |
| **Occupation** | | | |
| | Housewife | 13 | 8.6 |
| | Daily laborer | 46 | 30.5 |
| | Government employee | 12 | 7.9 |
| | Unemployed | 38 | 25.2 |
| | Other | 42 | 27.8 |
| **Diabetes Mellitus** | | | |
| | Yes | 14 | 9.3 |
| | No | 137 | 90.7 |
| **MDR-TB contact** | | | |
| | Yes | 7 | 4.6 |
| | No | 144 | 95.4 |
| **TB Category** | | | |
| | New cases | 137 | 90.7 |
| | Re-treatment | 14 | 9.3 |
| **Specimen type** | | | |
| | Abscess | 1 | 0.7 |
| | Ascetic Fluid | 5 | 3.3 |
| | CSF | 3 | 2.0 |
| | Lymph node aspirate | 99 | 65.6 |
| | Pleural Fluid | 32 | 12.2 |
| | Peritoneal fluid | 8 | 5.3 |
| | Pericardium fluid | 1 | 1.0 |
| | Pus | 2 | 2.0 |

**Table 2. Phenotyping drug resistance profiles for first-line drugs among confirmed EPTB patients by patient history profile, 2020.**

| Drug resistance pattern | Isolates from new cases, n (%) | Isolates from previously treated cases, n % | Total n (%) |
|---|---|---|---|
| Total tested Isolate | 137 | 14 | 151 |
| Susceptible | 115(83.9) | 7(50.0) | 122(80.8) |
| **Resistance to any drug** | | | |
| STM | 7(5.1) | 2(14.2) | 9 (6.0) |
| INH | 19(13.8) | 3(21.4) | 22 (14.6) |
| RIF | 11(8.0) | 3(21.4) | 14 (9.3) |
| EMB | 3(2.2) | - | 3 (2.0) |
| PZA | 19(13.9) | 3(21.4) | 22(14.6) |
| **Mono resistance** | | | |
| INH | 2(1.5) | 1(7.1) | 3 (2.0) |
| PZA | 8(5.8) | 1(7.1) | 9(5.9) |
| **Resistance to more than one drug** | | | |
| INH + RIF | 11(8.0) | 3(21.4) | 14(9.3) |
| INH + PZA | 12(8.6) | 3(21.4) | 15(9.9) |
| INH + STM | 7(5.1) | 2(14.3) | 9(6.0) |
| INH + EMB | 3(2.2) | - | 3(1.32) |
| INH + RIF + STM | 4(2.9) | 1(7.1) | 5(3.3) |
| INH + RIF + EMB | 2(1.5) | - | 2(1.3) |
| INH + RIF + PZA | 8(5.8) | 2(14.3) | 10(6.6) |
| INH + RIF + EMB + PZA | 1(0.7) | - | 1(0.7) |
| STM + INH + RIF + PZA | 4(2.9) | 1(7.1) | 5(3.3) |
| INH + RIF or MDR | 11(8.0) | 3(21.4) | 14 (9.3) |

STM-Streptomycin, INH-Isoniazid, RIF-Rifampicin, EMB-Ethambutol, PZA-Pyrazinamide, MDR-Multi drug resistance.

## Strain typing (spoligotyping)

All 151 MTB isolates were characterized by spoligotyping. Of 151 *MTB* isolates, 142(94.0%) displayed known patterns, while nine (6.0%) isolates not matched to the MIRU-VNTR*plus*database.

**Table 3. Drug resistance profiles for second-line drugs MDR-TB cases according to new cases and previously treated cases (n = 14), 2020.**

| Drug resistance pattern | Isolates from new cases, n (%) | Isolates from previously treated cases, n (%) | Total n (%) |
|---|---|---|---|
| **Total Isolate** | **11** | **3** | **14** |
| Amikacin | - | - | - |
| Capromycin | 1(9.1) | - | 1 (7.1) |
| Ethionamide | 1(9.1) | - | 1(7.1) |
| Kanamycin | - | - | - |
| Ofloxacin | 1(9.1) | 1(33.3) | 2 (14.3) |
| Moxifloxacin | 1(9.1) | 1(33.3) | 2 (14.3) |
| Pre-XDR | 2(18.2) | 1(33.3) | 3 (21.4) |

XDR-Extensively Drug-Resistant.

Accordingly, 41 different spoligotype patterns were identified and categorized under 11 families [Table 4]. The most predominant strain types observed were SIT149 33 (21.2%), SIT53 22 (14.6%), and SIT26 14 (9.6%). The most predominant lineages identified were T family 81 (53.6%), Central Asia Strain 29 (19.2%), Haarlem 15 (9.9%) and unknown 9 (6.0%) [Table 4].

## Discussion

In the current study, the molecular characteristics and drug resistance profiles of MTBC isolates from non-pulmonary sources were determined. The overall proportion of MDR-TB in new cases was 11 (8%) and 3 (21.4%) in previously treated cases. Of MDR-TB isolates, 3 (21.4%) were pre-XDR-TB. Two percent of isolates were INH mono-resistant. The molecular characteristics result showed that the T family was the most dominant 81 (53.6%) followed by Central Asia Strain 29 (19.2%) and Haarlem 15 (9.9%). The overall prevalence of *M.bovis* was 2% [3].

In the current study resistance to INH was the most frequent. This proportion is higher than the earlier result reported from Ethiopia [23] and Thailand [24]. However, our finding is similar to the results reported in India [25] and South Korea [26]. Another two studies reported from India indicated higher proportions of INH resistance (>30%) [27, 28] than our finding. In this study, INH mono-resistance was 3 (2%) which is relatively similar to the earlier study reported from Ethiopia [29] and India [27]. Another study reported from North India [30] indicated a higher proportion of INH mono-resistance than our result. This difference could be due to variation in treatment adherence, mutations, early introduction of drugs in the country, and high dose INH use for MDR treatment.

Resistance to PZA was also the most frequent in the current study. Although we could not find a previous study that determined the proportion of resistance against PZA in EPTB specimens, the study reported on both pulmonary TB and EPTB indicated similar results with our findings [31]. In contrast, a study reported from Myanmar [32] showed a high proportion of resistance against PZA in both pulmonary and EPTB specimens. The difference between previous findings and our result might be due to the burden of MDR-TB and treatment adherence levels in the study areas. Although the proportion of PZA resistance is high in MTBC cases, the phenotypic PZA susceptibility test is rarely performed due to technical difficulties. Currently, the MGIT 960 liquid culture method is the only WHO recommended methodology for PZA susceptibility testing, but this test is associated with a high rate of false-positive resistance results [33, 34]. Due to this limitation, the sequencing method is a more promising technique for the rapid and accurate detection of PZA resistance [34].

In our study, the proportion of MDR-TB among new and previously treated patients was 9.3% [14] in EPTB patients. Previous studies reported from India indicated a higher MDR-TB cases proportion than our findings [25, 27, 28]. In contrast, two studies from Korea [26, 35] one study from Thailand [24], and one study from Ethiopia [23] have reported a lower MDR-TB proportion than our findings. The possible reasons for the difference between our findings and previous studies results could be the non-adherence level, lost to follow up, poor drug supply chain management, and quality of drugs used [36]. Moreover, high MDR-TB in our study might be due to the difference in the study setting where the current study is conducted in an urban setting. However, our findings were consistent with previous studies [26, 28, 30, 37, 38].

The proportion of pre-XDR-TB cases among MDR-TB patients in the present study was 21.4% [3]. The finding of the current study is similar to the previous study reported from India in which the proportion of pre-XDR-TB among MDR-TB patients was 18.4% [30]. In contrast

**Table 4. Spoligotyping pattern, octal codes, SIT and lineage of extrapulmonary *M. tuberculosis* isolates (n = 151), May, 2020.**

| Lineage /Family | SIT | No. (%) | Octal Number | Spoligotype pattern |
|---|---|---|---|---|
| **Beijing** | 1 | 1(0.7) | 000000000003771 | (spoligotype pattern) |
| **X1** | 336 | 1(0.7) | 777776777760731 | (spoligotype pattern) |
| **Haarlem** | | **15(9.9)** | | |
| H1 | 47 | 1(0.7) | 777777774020771 | (spoligotype pattern) |
| H3 | 116 | 1(0.7) | 777767775720771 | (spoligotype pattern) |
| H3 | 121 | 1(0.7) | 777777775720771 | (spoligotype pattern) |
| H3 | 3 | 2(1.3) | 000000007720771 | (spoligotype pattern) |
| H3 | 50 | 2(1.3) | 777777777720771 | (spoligotype pattern) |
| H3 | 764 | 4(2.6) | 777757777720771 | (spoligotype pattern) |
| H4 | 777 | 4(2.6) | 777777777420771 | (spoligotype pattern) |
| **Bovis** | | **3(2.0)** | | |
| Bovis1 | 694 | 2(1.3) | 777776777760731 | (spoligotype pattern) |
| Bovis1 | 820 | 1(0.7) | 676763777777600 | (spoligotype pattern) |
| **T** | | **81(53.6)** | | |
| T1 | 1129 | 2(1.3) | 776777777760771 | (spoligotype pattern) |
| T1 | 131 | 9(6.0) | 777717777760771 | (spoligotype pattern) |
| T1 | 393 | 3(2.0) | 777757777760771 | (spoligotype pattern) |
| T1 | 3315 | 1(0.7) | 376777737760771 | (spoligotype pattern) |
| T1 | 53 | 22(14.6) | 777777777760771 | (spoligotype pattern) |
| T2 | 52 | 2(2.3) | 777777777760731 | (spoligotype pattern) |
| T2 | 875 | 2(1.3) | 777717777760731 | (spoligotype pattern) |
| T3 | 1745 | 1(0.7) | 773737777760771 | (spoligotype pattern) |
| T3 | 37 | 6(4.0) | 777737777760771 | (spoligotype pattern) |
| T3-ETH | 149 | 33(21.2) | 777000377760771 | (spoligotype pattern) |
| **Ural** | | **6(4.0)** | | |
| U | 1729 | 1(0.7) | 700000004177771 | (spoligotype pattern) |
| U | 910 | 4(2.6) | 700000007177771 | (spoligotype pattern) |
| U | 602 | 1(0.7) | 777777770000771 | (spoligotype pattern) |
| **Unknown strain** | | **9(6.0)** | | |
| Not defined | Not defined | 1(0.7) | 700000017774771 | (spoligotype pattern) |
| Not defined | Not defined | 1(0.7) | 777774777420771 | (spoligotype pattern) |
| Not defined | Not defined | 2(1.3) | 770000037760771 | (spoligotype pattern) |
| Not defined | Not defined | 1(0.7) | 777737377720771 | (spoligotype pattern) |
| Not defined | Not defined | 2(1.3) | 017777777760731 | (spoligotype pattern) |
| Not defined | Not defined | 2(1.3) | 777777747420771 | (spoligotype pattern) |
| **Latin American Mediterranean** | | **6(4.0)** | | |
| LAM9 | 1176 | 2(1.3) | 777757607760771 | (spoligotype pattern) |
| LAM9 | 42 | 3(2.0) | 777777607760771 | (spoligotype pattern) |
| LAM7 Tur | 41 | 1(0.7) | 777777404760771 | (spoligotype pattern) |
| **East African Indian** | | **1(0.7)** | | |
| EAI5 | 126 | 1(0.7) | 477777777413771 | (spoligotype pattern) |
| **Central Asia Strain** | | **29(19.2)** | | |
| CAS | 22 | 7(4.6) | 703777400001771 | (spoligotype pattern) |
| CAS1-Delhi | 25 | 2(1.3) | 703777740003171 | (spoligotype pattern) |
| CAS1-Delhi | 26 | 14(9.3) | 703777740003771 | (spoligotype pattern) |
| CAS1-Delhi | 289 | 3(2.0) | 703777740003571 | (spoligotype pattern) |
| CAS | 357 | 1(0.7) | 703777740000771 | (spoligotype pattern) |

(*Continued*)

**Table 4.** (Continued)

| Lineage /Family | SIT | No. (%) | Octal Number | Spoligotype pattern |
|---|---|---|---|---|
| CAS1-Delhi | 429 | 1(0.7) | 703777740003731 | ■■■□□□□■■■■■■■■■■■■■□□□□□□□□□□□■■■■□■■■ |
| CAS1-Delhi | 794 | 1(0.7) | 703757740003771 | ■■■□□□□■■■■■□■■■■■■■■□□□□□□□□□□□■■■■■■■■ |

CAS-Central Asia Strain, EAI-East African Indian, LAM-Latin American Mediterranean, ETH-Ethiopia, H-Haarlem, SIT-Spoligotype International Type.

to our findings, a study reported from India indicated a high proportion (38.2%) of pre-XDR-TB among MDR-TB patients [39]. However, a lower proportion was reported in a similar study conducted in Pakistan [40]. This difference most probably occurred due to treatment failure and MDR-TB burden in the study areas, treatment non-adherence, quality of the drug used, and, TB program-related issues.

In the current study, different strain types of MTB; Beijing, X1, Haarlem, Bovis, T, Ethiopia, Ural, Central Asia Strain, East African-Indian, and Latin American-Mediterranean, were identified. There were also some orphan strains assigned to their most appropriate lineage and sub-lineage using the TB insight database. The most frequent of lineages in the current study were T, CAS, and H. The most predominant strain spoligotypes were SIT149, SIT53, and SIT26. These results are in line with two earlier studies reported from Italy [41] and Ethiopia [42]. A study reported from Addis Ababa has also shown a high proportion of SIT53 strain and CAS family strains [41]. The number of SIT149 isolates in our study was thirty-two isolates higher than the number registered in the SITVIT2 international database. This might be due to the association of this familiar strain with East African countries, especially in Ethiopia. Besides, Addis Ababa is the capital city of Ethiopia and the place for the African Union with a high inflow of individuals that might be the possible reason for the presence of genetically diversified MTBC strain in the present study.

In the current study, only 3 (2%) of *M. bovis* were identified among MTBC isolates from EPTB patients. A similar finding has been reported in previous studies of Ethiopia [23], Mozambique [43], Madagascar [44], and India [45]. The low proportion of *M. bovis* in the present study could be due to the fact that the majority of the participants were urban dwellers which are not associated with livestock.

Our study has some limitations. As the two studies from which the isolates used in the current study did not target presumptive MDR EPTB patients, the finding of the present study cannot be generalized for the second-line drug resistance. Moreover, this study was included participants living in Addis Ababa which could not reflect the actual molecular characteristics and drug resistance patterns in EPTB patients in the country. Besides, the molecular characterization of *Mycobacterial tuberculosis* strains was not performed using advanced molecular diagnostic tools such as mycobacterial interspersed repetitive unit variable number tandem repeat (MIRU-VNTR) and sequencing other than spoligotyping. However, our findings provide important evidence on molecular characteristics and drug resistance patterns of EPTB.

## Conclusion

The present study showed different strain types of *MTB* circulating in Addis Ababa. T family and Central Asia Strains are the most dominant lineages and SIT149 was the most predominant strain. In the current study, high proportions of MDR-TB and pre-XDR-TB were identified in patients with EPTB. Strengthen MDR-TB and pre-XDR-TB prevention is required to combat these strains in EPTB patients. Further large scale study is recommended to provide comprehensive molecular characteristics and drug resistance patterns in EPTB patients.

## Supporting information

**S1 File.**
(SAV)

## Acknowledgments

The authors would like to acknowledge the Ethiopian Public Health Institute and Addis Ababa University for providing materials and facilities during this study. The authors are also grateful for the study participants whose specimens and data were used in this study.

## Author Contributions

**Conceptualization:** Getu Diriba, Abebaw Kebede, Habteyes Hailu Tola, Dessie Abera, Kassu Desta.

**Data curation:** Getu Diriba, Abebaw Kebede, Habteyes Hailu Tola, Bazezew Yenew, Shewki Moga, Desalegn Addise, Ayinalem Alemu, Zemedu Mohammed, Muluwork Getahun, Mengistu Fantahun, Mengistu Tadesse, Biniyam Dagne, Misikir Amare, Gebeyehu Assefa, Dessie Abera, Kassu Desta.

**Formal analysis:** Getu Diriba, Bazezew Yenew, Shewki Moga, Desalegn Addise, Ayinalem Alemu, Zemedu Mohammed, Mengistu Tadesse, Biniyam Dagne, Misikir Amare.

**Funding acquisition:** Getu Diriba, Abebaw Kebede, Muluwork Getahun.

**Investigation:** Getu Diriba.

**Methodology:** Getu Diriba, Abebaw Kebede, Habteyes Hailu Tola, Zemedu Mohammed, Dessie Abera, Kassu Desta.

**Project administration:** Getu Diriba, Bazezew Yenew.

**Resources:** Getu Diriba.

**Software:** Getu Diriba, Zemedu Mohammed.

**Supervision:** Getu Diriba.

**Validation:** Getu Diriba, Abebaw Kebede, Habteyes Hailu Tola.

**Visualization:** Getu Diriba.

**Writing – original draft:** Getu Diriba.

**Writing – review & editing:** Getu Diriba, Abebaw Kebede, Habteyes Hailu Tola, Kassu Desta.

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
