## [Decision Letter · Decision Letter 0]

28 Jul 2020

PONE-D-20-17590

Molecular Characterization and Drug Resistance Patterns of Mycobacterium Tuberculosis Complex in Extrapulmonary Tuberculosis Patients in Addis Ababa, Ethiopia

PLOS ONE

Dear Dr. Diriba,

Thank you for submitting your manuscript to PLOS ONE. After careful consideration, we feel that it has merit but does not fully meet PLOS ONE’s publication criteria as it currently stands. Therefore, we invite you to submit a revised version of the manuscript that addresses the points raised during the review process.

We look forward to receiving your revised manuscript.

Kind regards,

Shampa Anupurba, MD

Academic Editor

PLOS ONE

Journal Requirements:

2. Please provide additional details regarding participant consent. In the ethics statement in the Methods and online submission information, please ensure that you have specified (1) whether consent was informed and (2) what type you obtained (for instance, written or verbal, and if verbal, how it was documented and witnessed). If the need for consent was waived by the ethics committee, please include this information.

Additional Editor Comments (if provided):

The language has to be substantially modified in order to be comprehensible

Reviewers' comments:

Reviewer's Responses to Questions

**Comments to the Author**

1. Is the manuscript technically sound, and do the data support the conclusions?

Reviewer #1: Partly

Reviewer #2: Yes

2. Has the statistical analysis been performed appropriately and rigorously? 

Reviewer #1: N/A

Reviewer #2: Yes

3. Have the authors made all data underlying the findings in their manuscript fully available?

Reviewer #1: Yes

Reviewer #2: Yes

4. Is the manuscript presented in an intelligible fashion and written in standard English?

Reviewer #1: No

Reviewer #2: No

5. Review Comments to the Author

Reviewer #1: Manuscript by Diriba et al reports on the characteristics of M.tuberculosis strains isolated from extrapulmonary TB patients residing in Addis Ababa (Ethiopia). Specifically, phenotypic drug susceptibility patterns and spoligotypes of 151 MTBC strains were analysed and reported.

Subject of the study is of interest as Ethiopia still very much remains a white spot in TB world phylogeography while drug resistance patterns of MTBC strains circulating there are still poorly understood especially in extrapulmonary TB isolates.

That said, in my opinion, the manuscript has a number of serious problems which need to be fixed before it can be considered for publicatrion.

First of all, quality of written English is poor and certain fragments are not fully comprehensible. This applies predominantly to Introduction and Discussion section. Grammar is key problem (passive/active voice; tenses etc). Text should be proof read, ideally by a native speaker, and revised/corrected accordingly.

Content wise, study design is a key problem. Authors refer to one published and one unpublished study when describing study settings and patient population, but it's far from clear how many specimens were taken from what study, how patients were enrolled and specimens collected (consecutive specimens? convenience sample? individial vs serial isolates? inclusion criteria? etc). Having read the paper authors were referring to (Fantahun et al., PLoS ONE 2019) i realised it included 152 specimens and nearly all the clinical, epidemiological and demographic characteristics were almost the same (current study included 151 specimens), so it's worth expanding and clarifying on whether exactly the same selection of isolates was tested in the current study.

Depending on representativeness of the study population (which should be clarified as discussed above), all the conclusions and considerations with regards to prevalence of specific drug resistances, spoligotypes, and other characteristics vs other Ethiopian studies and reports from other countries should be adjusted. If this collection was essentially a convenience sample, no firm conclusions regarding prevalence could be made.

Another problem is hypothesis and principal aim of the study. What was the reason for genotyping and how exactly this type of knowledge can contribute to TB management in Ethiopia especially (as authors rightly state) EPTB is not as transmissible? I suggest to put this study into a broader context of TB/MTBC phylogeography; to this end use of another database (VNTR-MIRUplus) as an alternative or compelentary to SITVIT could be recommended.

Other points:

- author state that "Lymph node TB was the most common site involved..." (Page 9, Line 211-212). However, this will be heavily influenced by inclusion criteria (see above) - if only patients suspected of having Lymph node impairment have been enrolled, than this is not a surprising finding. As I say, defining inclusion criteria will help significantly.

- PZA sensitivity: authors rightly state that reproducibility of phenotypic PZA testing is not ideal. It's worth referring to the recent WHO guidelines (eg 2018) where sequencing is recommended as a reference method.

- Both Background and Discussion sections, as well as fragments of Materials and Methods are far too long and should be shortened. In Introduction. authors should concentrate on their hypothesis and what is known so far; in Discussion messages should be more concise. Pages 10-12 are very difficult to read as they are overloaded with non-relevant information, this should be significantly shortened. In Methods (Page 6) a simple reference to Spoligotyping methodology (eg Kamerbeek et al) will suffice, no need to describe it in details.

Reviewer #2: The molecular typing of EPTB isolates (which is often neglected by programme managers) from Ethiopia is appreciated. The samples size is also not very large but acceptable. The findings of M. bovis in about 2% a important finding. However, the study has following concerns and these need to be addressed.

1. The language needs significant improvisation. The Introduction is not very focused on the need of this study. Some of the studies cited in the introduction do not find place in discussion at all, which have clear message and the excerpts should have been cited in discussion as well.

2.Like Introduction the discussion is diffuse and lacks a flow. It has a number of repetitions, and irrelevant statement.

3. The major limitation of the study is that conclusions are based on Spoligotyping. We know that spoligotypic classification lacks discriminatory power and therefore, MIRU-VNTR must have been done. However, considering that this study is coming from Ethiopia, I would not insist on this aspect, but the authors must create a separate para on the limitation of this study, wherein why MIRU-VNTR could not be done, must be mentioned.

6. PLOS authors have the option to publish the peer review history of their article (what does this mean?). If published, this will include your full peer review and any attached files.

Reviewer #1: **Yes: **Vladyslav Nikolayevskyy

Reviewer #2: **Yes: **Prof. Sarman Singh

---

## [Author Response · Author response to Decision Letter 0]

10 Sep 2020

Dear Editor,

Thank you very much for your informative comments on our manuscript entitled “Molecular Characterization and Drug Resistance Patterns of Mycobacterium tuberculosis Complex in Extrapulmonary Tuberculosis Patients in Addis Ababa, Ethiopia”. We have addressed your comments one by one. We also appreciate you for allowing us to revise our manuscript and correct errors in the previous version. We thank the reviewers for their informative comments, and our point-by-point responses to the reviewers’ comments are given below. Also, we would like to inform you that we have used highlights to indicate where we made changes in response to the reviewers’ comments. 

Review Comments to the Author

Reviewer #1: Manuscript by Diriba et al reports on the characteristics of M.tuberculosis strains isolated from extrapulmonary TB patients residing in Addis Ababa (Ethiopia). Specifically, phenotypic drug susceptibility patterns and spoligotypes of 151 MTBC strains were analysed and reported. Subject of the study is of interest as Ethiopia still very much remains a white spot in TB world phylogeography while drug resistance patterns of MTBC strains circulating there are still poorly understood especially in extrapulmonary TB isolates.That said, in my opinion, the manuscript has a number of serious problems which need to be fixed before it can be considered for publicatrion.

1. First of all, quality of written English is poor and certain fragments are not fully comprehensible. This applies predominantly to Introduction and Discussion section. Grammar is key problem (passive/active voice; tenses etc). Text should be proof read, ideally by a native speaker, and revised/corrected accordingly.

Response: Thank you very much for your informative comments. We have addressed you comments across the manuscript. Errors related to language usage and grammer have revised by native English speaker.

2. Content wise, study design is a key problem. Authors refer to one published and one unpublished study when describing study settings and patient population, but it's far from clear how many specimens were taken from what study, how patients were enrolled and specimens collected (consecutive specimens? convenience sample? individial vs serial isolates? inclusion criteria? etc). Having read the paper authors were referring to (Fantahun et al., PLoS ONE 2019) i realised it included 152 specimens and nearly all the clinical, epidemiological and demographic characteristics were almost the same (current study included 151 specimens), so it's worth expanding and clarifying on whether exactly the same selection of isolates was tested in the current study.

Response: Thank you for the pertinent comment. To address this comment we have described the process of sampling and how many isolates were taken from the previously published study and unpulished study as follows

The current study was conducted on clinical isolates of MTBC from two previous EPTB studies conducted in Addis Ababa, Ethiopia. One of the survey data used in the current study was published and the study participants were 152 patients with EPTB (17) while the second survey which is not published was enrolled 778 patients. Sixty eight EPTB presumptive patients were MTBC positive out of 152 (from the first survey), whereas 85 were positive from 778 patients (from second survey). In total, 153(16.5%) MTBCs isolates were obtained from the total of 930 patients from the two previous studies. Since two isolates were not recovered by sub-culturing, 151 MTBC isolates were used in the present study. In both studies, patients were enrolled consecutively from upon the arrival of the participants. In the previous studies from which isolates used in the current study, single sample per patient was collected and tested for TB using culture and Xpert MTB/RIF assay. 

3. Depending on representativeness of the study population (which should be clarified as discussed above), all the conclusions and considerations with regards to prevalence of specific drug resistances, spoligotypes, and other characteristics vs other Ethiopian studies and reports from other countries should be adjusted. If this collection was essentially a convenience sample, no firm conclusions regarding prevalence could be made.

Response: Thank you for the critical observation. We have retrieved MTBC clinical isolates from the previous two EPTB studies. We believe these two studies were based on statistically determined sample size and the sampling the study participants are represented of their respective study population. This study mainly aimed to document the genotypic and phenotypic charactersitics of the isolates from the two studies. Thus, there are clear difference between previous studies and our study aims. 

4. Another problem is hypothesis and principal aim of the study. What was the reason for genotyping and how exactly this type of knowledge can contribute to TB management in Ethiopia especially (as authors rightly state) EPTB is not as transmissible? I suggest to put this study into a broader context of TB/MTBC phylogeography; to this end use of another database (VNTR-MIRUplus) as an alternative or compelentary to SITVIT could be recommended.

Response: 

Thank you very much for your crtical observation. We have addressed your comment by adding the reasons why gewnotyping method was used and the value of this technique for the TB management in Ethiopia (line 78-80). Since we did not conduct VNTR-MIRUplus due to lack of the technique in our setup, we could not compare our result with the indicated database as complimentary to SITVIT databse. To address this comment we have added a sentence in the limitation part of this study. 

5- Other points: author state that "Lymph node TB was the most common site involved..." (Page 9, Line 211-212). However, this will be heavily influenced by inclusion criteria (see above) - if only patients suspected of having Lymph node impairment have been enrolled, than this is not a surprising finding. As I say, defining inclusion criteria will help significantly.

Response: We have indicated as "Lymph node TB was the most common site involved..." just to give good insight about the source of the MTBC isolates in detail. We have removed this sentence from the discaussion parts to address your comment. 

6- PZA sensitivity: authors rightly state that reproducibility of phenotypic PZA testing is not ideal. It's worth referring to the recent WHO guidelines (eg 2018) where sequencing is recommended as a reference method.

Response: Corrected accordingly to the comment. We have indicated in the discussion section as WHO has recently recommends sequencing as a reference method (Line 196-201)

7- Both Background and Discussion sections, as well as fragments of Materials and Methods are far too long and should be shortened. In Introduction. authors should concentrate on their hypothesis and what is known so far; in Discussion messages should be more concise. Pages 10-12 are very difficult to read as they are overloaded with non-relevant information, this should be significantly shortened. In Methods (Page 6) a simple reference to Spoligotyping methodology (eg Kamerbeek et al) will suffice, no need to describe it in details.

Response: Thank you for the pertinent comment and suggestion. We have corrected all errors based on your comment.

Reviewer reports: #2

Reviewer #2: The molecular typing of EPTB isolates (which is often neglected by programme managers) from Ethiopia is appreciated. The samples size is also not very large but acceptable. The findings of M. bovis in about 2% a important finding. However, the study has following concerns and these need to be addressed.

1. The language needs significant improvisation. The Introduction is not very focused on the need of this study. Some of the studies cited in the introduction do not find place in discussion at all, which have clear message and the excerpts should have been cited in discussion as well.

Response: Thank you for the informative comments and suggestions. We have revised the introduction to focus to our study aobjective. We have revised the introduction and discussion section based on your comment to clarify and focuse.

2. Like Introduction the discussion is diffuse and lacks a flow. It has a number of repetitions, and irrelevant statement.

Response: We removed inrrelevant statements and revised the introduction and discussion sections in detail. 

3. The major limitation of the study is that conclusions are based on Spoligotyping. We know that spoligotypic classification lacks discriminatory power and therefore, MIRU-VNTR must have been done. However, considering that this study is coming from Ethiopia, I would not insist on this aspect, but the authors must create a separate para on the limitation of this study, wherein why MIRU-VNTR could not be done, must be mentioned.

Response: Thank you very much for the suggestion. We did not conduct advanced molecular tests such as MIRU-VTRN and sequencing due to lack of the setup. To address this comment, we have included an explanation in the limitation part of the revised manuscript (Line 241-244).

Yours, Sincerely 

Getu Diriba

---

## [Decision Letter · Decision Letter 1]

5 Oct 2020

PONE-D-20-17590R1

Molecular Characterization and Drug Resistance Patterns of Mycobacterium Tuberculosis Complex in Extrapulmonary Tuberculosis Patients in Addis Ababa, Ethiopia

PLOS ONE

Dear Dr. Diriba,

Thank you for submitting your manuscript to PLOS ONE. After careful consideration, we feel that it has merit but does not fully meet PLOS ONE’s publication criteria as it currently stands. Therefore, we invite you to submit a revised version of the manuscript that addresses the points raised during the review process.

The reviewers have gone through your manuscript thoroughly and given their comments.Please try to improve it further. Much improvement is needed in the discussion section. Moreover, the English is still poor with grammatical mistakes and mixing of tenses. Apart from their suggestions, I have tried to correct the language although I would still suggest that a native speaking person should be consulted.Further as suggested, it is advised that MIRU-VNTR database be used.

Line 33- 'strain' instead of strains

Line 46- 'strains were' instead of strain was

Line 63- Insert 'is' between disease and manifested

Line 64- Delete 'EPTB occurs in any part of the body except for the lungs'. 'Replace 'It becomes' with 'EPTB is'

Line 68- 'are' instead of 'become'

Lines 71-72- Can be rewritten as 'EPTB specimens have several inhibitors (...) that compromise the quality of PCR amplification leading to low sensitivity and false-negative results'.

Line 74- Delete 'remaining'

Line 79- Insert 'are' between 'that'and 'circulating'

Line 81-Delete 'Even though EPTB is a public health problem in Ethiopia' and start with 'There are a few studies that have reported .....

Line 95- Delete 'was' after study

Line 96- Delete ''was' after published and 'in' after enrolled

Line 99 Delete 's' after MTBC

Line 101- Delete 'from' after consecutively

Line 108- pyrazinamide instead of Pyrazinamide

Line 111- RFP should be replaced by RIF and rephrase the sentence

Line 113- 'The bacterial inocula...'

Line 114- '0.5 ml of 1:100 dilution was added into tube containing no drug'

Line 115- 'every hour' and delete one after every

Line 122-124-All names of drugs should be in lower case. Also moxifloxacin (MOX) should be added.

Line 125- 'strain identification' instead of strains

Line 127- Replace 'The complexes that identified are' with 'which include...'

Line 133- Replace 'however' with 'and'

Line 135- Replace 'base' by 'basis'

Line 146- Delete 'Since' and write 'isolates' instead of isolate

Line 155- '99(65.6%)had tuberculous lymphadenitis while 12.2% had pleural. If total number of lymph node aspirates were maximum, then the number of TB lymphadenitis would also be highest. This should be checked. Also, Table 1 can be deleted.

Line 159- Delete 'of'

Line 160- Replace 'list' with 'least', delete 'of'

Line 161-'The proportion of MDR-TB was 9.3%, which included 3 of 14 (21.4%) previously treated and 11 of 137 (8%) newly treated cases.

Line 167- Delete 'developed'

Lines 167-168- One of the three (33.3%) previously treated and two of eleven (18.2%) new cases were found to be pre XDR cases.

Line 171-...three (2.0%) 'were' M. bovis

Line 172- Delete entire line

Line 175- Delete 'were'

Line 176- 'Of those isolates their patterns are known...' may be rephrased as 'The isolates with known patterns...'

Line 184- Replace prevalence by 'proportion'

Line 185- Replace 21.14% by 21.4%. Delete 'developed'

Table 2 - In last row under column previously treated correct 21.14% to 21.4%

Line 188- Delete 'which'

Line 202- Replace 'the' by 'a' and shown by 'showed'

Line 211- 9.3% instead of high

Line 213- Delete 'the previous'

Line 218- Insert 'the fact that ' after due to and omit comma

Line 219- Delete 'findings' after studies.

Line 220- Delete 'high'

Line 221- Delete 'is'

Line 224- Replace the by 'a'

Lines 225-227- Similar explanation for disparity of results are repeatedly used throughout Discussion which should be avoided. Instead authors may first mention similarities and differences between their findings and findings from other studies and mention the probable reason in the end.

Line 231- Replace 'prevalent' by 'frequent'

Line 234- 'has' instead of is

Lines 238-240- Rephrase, also check 'activates'

Line 244- Insert 'fact that' after due to the

Line 247- 'study did not target presumptive MDR EPTB patients', Compare with Line 218. Proper selection criteria should be mentioned under Materials and Methods before drawing inferences.

Line 252- MIRU-VNTR should be expanded

Reference 22 should be corrected

We look forward to receiving your revised manuscript.

Kind regards,

Shampa Anupurba, MD

Academic Editor

PLOS ONE

Reviewers' comments:

Reviewer's Responses to Questions

**Comments to the Author**

1. If the authors have adequately addressed your comments raised in a previous round of review and you feel that this manuscript is now acceptable for publication, you may indicate that here to bypass the “Comments to the Author” section, enter your conflict of interest statement in the “Confidential to Editor” section, and submit your "Accept" recommendation.

Reviewer #1: (No Response)

Reviewer #2: (No Response)

2. Is the manuscript technically sound, and do the data support the conclusions?

Reviewer #1: Partly

Reviewer #2: Partly

3. Has the statistical analysis been performed appropriately and rigorously? 

Reviewer #1: N/A

Reviewer #2: (No Response)

4. Have the authors made all data underlying the findings in their manuscript fully available?

Reviewer #1: Yes

Reviewer #2: No

5. Is the manuscript presented in an intelligible fashion and written in standard English?

Reviewer #1: No

Reviewer #2: No

6. Review Comments to the Author

Reviewer #1: I would like to thank Authors for addressing most of the Reviewer's comments which in my opinion helped to improve the manuscript and make conclusions better supported by the data also highlighting the study limitations.

However there is a number of points that need to be addressed before the manuscript could be accepted for publication:

- although Authors state the manuscript has been reviewed by a native English speaker, problems with English grammar persist. There are multiple errors and inaccuracies mainly in the Abstract and Introduction and also methods which, if remain uncorrected, will affect reader's experience and understanding. Examples include:

- Line 49 "The strain was mostly grouped into SIT149, SIT53 and SIT26" (should be "strainS");

- Lines 49-50 (again strainS i.e. plural but not strain single);

- Lines 71-73 - Materials (not assays) have inhibitors

- Lines 95-97 "One of the survey data used in the current study was published 152 patients with EPTB (17) while the second survey which is not published was enrolled 778 patients - problems with passive/active voices

- etc.

I strongly recommend that text again is revised, double checked and multiple inaccuracies corrected.

Factual errors/inaccuracies:

- Lines 149-150; 129 plus 29 is equal to 158 but a total number of strains was 151; please correct and re-calculate proportions if necessary;

- Line 181 and 190: please re-phrase as INH and PZA resistance prevalence was the same.

- Line 211-212: please avoid using "high" in relation to pre-XDR proportion as absolute numbers are very low (3 out of 14). Please re-phrase to avoid misleading.

Other points:

- Lines 174-180 and whole Discussion section; please provide absolute numbers (not only proportions) as numbers are generally low and presenting proportions alone is misleading.

- I fully appreciate what authors are saying about lack of resources and inability to do MIRU-VNTR typing. However my suggestion regarding MIRU-VNTRplus database was not about performing VNTR; i recommended using this resource to re-classify strains using novel (and widely recognised) nomenclature. It's up to the Editor to decide whether it's essential; personally i think using more advanced nomenclature provided by MIRU-VNTRplus will make conclusions stronger and, importantly, more generalisable .

Reviewer #2: I regret to mention that manuscript is not yet publishable. Several concerns have not yet been addressed despite very positive suggestions.

For example "Line 81 to: Even though EPTB is a public health problem in Ethiopia, few studies have been reported on the genetic variability and drug resistance patterns of MTB in patients with EPTB in Addis Ababa (13, 14). Besides, there is no strong EPTB surveillance data on genetic diversity" How the EPTB is a public health problem? Usually only PTB is. So authors should provide some literature to make this statement. Also in these lines they mention "no strong EPTb surveillance data" ? What this means, authors need to explain. This study is about the molecular characterization of isolate. Which is clear from the isolates they included from two previous studies. Indeed this makes me puzzled. Why authors are again repeating the methods like staining and culture and single sample or otherwise. This study could have been straight if they were starting from the isolates recovered from previous two studies and they could have referred the previous studies.

2. I also it irrelevant and difficult to assimilate how they categorize the clinical details of all patients while including only number of Mycobacterial isolates (Table 1). This table is irrelevant.

There are several other issues in the study, especially the design and interpretation. Indeed this manuscript has very little chance to improve with suggestions and should be rejected. However, it is up to the editor to take decision.

7. PLOS authors have the option to publish the peer review history of their article (what does this mean?). If published, this will include your full peer review and any attached files.

Reviewer #1: No

Reviewer #2: **Yes: **Prof Sarman Singh

---

## [Author Response · Author response to Decision Letter 1]

19 Oct 2020

Cover letter for revised manuscript number: PONE-D-20-17590R2

Title: Molecular Characterization and Drug Resistance Patterns of Mycobacterium Tuberculosis Complex in Extrapulmonary Tuberculosis Patients in Addis Ababa, Ethiopia.

Revisions based on the comments, and questions of the editor and the reviewers

Editor

The reviewers have gone through your manuscript thoroughly and given their comments. Please try to improve it further. Much improvement is needed in the discussion section. Moreover, the English is still poor with grammatical mistakes and mixing of tenses. Apart from their suggestions, I have tried to correct the language although I would still suggest that a native speaking person should be consulted. Further as suggested, it is advised that MIRU-VNTR database be used.

Response: Thank you for the constructive comments and suggestions. The revised manuscript is language checked with a native speaker and based on your and the reviewers’ suggestion we applied the MIRU-VNTRplus database at this stage. Also, the discussions section is revised accordingly based on the comments. Besides the grammatical issues and the words were revised based on the editor suggestions and comments. Sentences are rephrased based on the given directions, and we track changed all the corrections in the revised manuscript. 

 Reviewer #1: 

Although Authors state the manuscript has been reviewed by a native English speaker, problems with English grammar persist. There are multiple errors and inaccuracies mainly in the Abstract and Introduction and also methods which, if remain uncorrected, will affect reader's experience and understanding. Examples include:

- Line 49 "The strain was mostly grouped into SIT149, SIT53 and SIT26" (should be "strainS");

- Lines 49-50 (again strainS i.e. plural but not strain single);

- Lines 71-73 - Materials (not assays) have inhibitors

- Lines 95-97 "One of the survey data used in the current study was published 152 patients with EPTB (17) while the second survey which is not published was enrolled 778 patients - problems with passive/active voices

- etc.

I strongly recommend that text again is revised, double checked and multiple inaccuracies corrected.

Response: Thank you for the valuable comments and suggestions. The current manuscript is language checked with a native speaker 

Factual errors/inaccuracies:

- Lines 149-150; 129 plus 29 is equal to 158 but a total number of strains was 151; please correct and re-calculate proportions if necessary;

Response: Thank you for the comment, it is corrected now. 

- Line 181 and 190: please re-phrase as INH and PZA resistance prevalence was the same.

Response: Thank you for the comment, we re-phrased it accordingly.

- Line 211-212: please avoid using "high" in relation to pre-XDR proportion as absolute numbers are very low (3 out of 14). Please re-phrase to avoid misleading.

Response: Thank you for the comment, we revised it accordingly.

- Lines 174-180 and whole Discussion section; please provide absolute numbers (not only proportions) as numbers are generally low and presenting proportions alone is misleading.

Response: Thank you for the comment; we revised using the absolute numbers based on the suggestion you given.

- I fully appreciate what authors are saying about lack of resources and inability to do MIRU-VNTR typing. However my suggestion regarding MIRU-VNTRplus database was not about performing VNTR; i recommended using this resource to re-classify strains using novel (and widely recognised) nomenclature. It's up to the Editor to decide whether it's essential; personally i think using more advanced nomenclature provided by MIRU-VNTRplus will make conclusions stronger and, importantly, more generalisable .

Response: Thank you for the valuable suggestion, we applied the MIRU-VNTRplus database and we revised the manuscript accordingly. 

Reviewer #2: 

I regret to mention that manuscript is not yet publishable. Several concerns have not yet been addressed despite very positive suggestions.

For example "Line 81 to: Even though EPTB is a public health problem in Ethiopia, few studies have been reported on the genetic variability and drug resistance patterns of MTB in patients with EPTB in Addis Ababa (13, 14). Besides, there is no strong EPTB surveillance data on genetic diversity" How the EPTB is a public health problem? Usually only PTB is. So authors should provide some literature to make this statement. Also in these lines they mention "no strong EPTb surveillance data" ? What this means, authors need to explain. This study is about the molecular characterization of isolate. Which is clear from the isolates they included from two previous studies. Indeed this makes me puzzled. Why authors are again repeating the methods like staining and culture and single sample or otherwise. This study could have been straight if they were starting from the isolates recovered from previous two studies and they could have referred the previous studies.

Response: Thank you for the comments and the concerns. We wrote the sentence “EPTB is a public health problem in Ethiopia” because it accounts for 31% of the TB cases in the country. But, we removed it in the revised manuscript based on the given comment. Since there is limited data on the circulating strains that are causing EPTB in the country including the current study setting, we understand it is a research gap. Besides, strain characterization is not routinely followed in the country. Thus, identifying the circulating MTBC strains causing EPTB is important. The current study is conducted using isolates recovered from previously conducted two studies in the study setting. We wrote the sentence “In the previous studies from which the isolates were used in the current study, a single sample per patient was collected and tested for TB using culture and Xpert MTB/RIF assay” to make clear how EPTB was ruled out. In the revised manuscript we remove this sentence. 

2. I also it irrelevant and difficult to assimilate how they categorize the clinical details of all patients while including only number of Mycobacterial isolates (Table 1). This table is irrelevant.

There are several other issues in the study, especially the design and interpretation. Indeed this manuscript has very little chance to improve with suggestions and should be rejected. However, it is up to the editor to take decision. 

Response: Thank you for the comment. We considered that it would be important if the demographic and clinical details of EPTB patients are described. For example, among patients with known HIV status, 26.2 % were HIV positive and most (65.6%) of the isolates were isolated from LN samples.

---

## [Decision Letter · Decision Letter 2]

23 Nov 2020

Molecular Characterization and Drug Resistance Patterns of Mycobacterium Tuberculosis Complex in Extrapulmonary Tuberculosis Patients in Addis Ababa, Ethiopia

PONE-D-20-17590R2

Dear Dr. Diriba,

We’re pleased to inform you that your manuscript has been judged scientifically suitable for publication and will be formally accepted for publication once it meets all outstanding technical requirements.

Kind regards,

Shampa Anupurba, MD

Academic Editor

PLOS ONE

Reviewers' comments:

Reviewer's Responses to Questions

**Comments to the Author**

1. If the authors have adequately addressed your comments raised in a previous round of review and you feel that this manuscript is now acceptable for publication, you may indicate that here to bypass the “Comments to the Author” section, enter your conflict of interest statement in the “Confidential to Editor” section, and submit your "Accept" recommendation.

Reviewer #1: All comments have been addressed

2. Is the manuscript technically sound, and do the data support the conclusions?

Reviewer #1: Partly

3. Has the statistical analysis been performed appropriately and rigorously? 

Reviewer #1: N/A

4. Have the authors made all data underlying the findings in their manuscript fully available?

Reviewer #1: Yes

5. Is the manuscript presented in an intelligible fashion and written in standard English?

Reviewer #1: Yes

6. Review Comments to the Author

Reviewer #1: I would like to thank the Authors for addressing the reviewer's comments and consider manuscript acceptable in its current format

7. PLOS authors have the option to publish the peer review history of their article (what does this mean?). If published, this will include your full peer review and any attached files.

Reviewer #1: No

---

## [Editor Report · Acceptance letter]

26 Nov 2020

PONE-D-20-17590R2 

Molecular Characterization and Drug Resistance Patterns of *Mycobacterium tuberculosis* Complex in Extrapulmonary Tuberculosis Patients in Addis Ababa, Ethiopia 

Dear Dr. Diriba:

I'm pleased to inform you that your manuscript has been deemed suitable for publication in PLOS ONE. Congratulations! Your manuscript is now with our production department. 

Kind regards, 

on behalf of

Dr. Shampa Anupurba 

Academic Editor

PLOS ONE